# Bipedal Stepping Controller Design Considering Model Uncertainty: A Data-Driven Perspective

**DOI:** 10.3390/biomimetics9110681

**Published:** 2024-11-07

**Authors:** Chao Song, Xizhe Zang, Boyang Chen, Shuai Heng, Changle Li, Yanhe Zhu, Jie Zhao

**Affiliations:** State Key Laboratory of Robotics and System, Harbin Institute of Technology, Harbin 150080, China; heng13514479054@163.com (S.H.); lichangle@hit.edu.cn (C.L.); yhzhu@hit.edu.cn (Y.Z.); jzhao@hit.edu.cn (J.Z.)

**Keywords:** model uncertainty, robust control, data-driven control, bipedal locomotion

## Abstract

This article introduces a novel perspective on designing a stepping controller for bipedal robots. Typically, designing a state-feedback controller to stabilize a bipedal robot to a periodic orbit of step-to-step (S2S) dynamics based on a reduced-order model (ROM) can achieve stable walking. However, the model discrepancies between the ROM and the full-order dynamic system are often ignored. We introduce the latest results from behavioral systems theory by directly constructing a robust stepping controller using input-state data collected during flat-ground walking with a nominal controller in the simulation. The model uncertainty discrepancies are equivalently represented as bounded noise and over-approximated by bounded energy ellipsoids. We conducted extensive walking experiments in a simulation on a 22-degrees-of-freedom small humanoid robot, verifying that it demonstrates superior robustness in handling uncertain loads, various sloped terrains, and push recovery compared to the nominal S2S controller.

## 1. Introduction

Reduced-order models have always played a crucial role in generating gait for humanoid robots. They capture the key physical characteristics of walking in humanoid robots, and due to their low-dimensional state and control variables, they can typically be computed in real time. Linear inverted pendulum (LIP) models are commonly used to generate feasible foot placements and have been widely studied in the robotics community over the past few decades [1,2]. The restrictive assumptions of LIP, such as fixed height and the neglect of angular momentum, lead to varied degrees of deviation in robots with different mass distributions. A notable issue is how to incorporate this model uncertainty into the design of the stepping controller and improve its robustness for walking.

The early LIP models are closer to quasi-static locomotion, requiring the center of pressure to remain within the convex hull formed by the foot contacts. More dynamic locomotion benefits from the introduce of the capturability theory and the viability theory, leading to the development of footstep planning based on the divergent component of motion (DCM) [3]. The idea of the DCM-based controller is to introduce constraints for the unstable components of LIP dynamics, thereby stabilizing the overall dynamics. The problem with this approach is that the LIP model and the constraints formulated for it both deviate from the original system. A possible direction is to use less conservative robust model predictive control (MPC) [4] by considering the effect of future deviation accumulation through the concept of constraint tightening. More precise reduced-order models (ROMs) include centroidal dynamics [5], and its simplified version, the single-rigid-body dynamics model, has shown impressive performance in bipedal robots [6], but they focus on solving for reference contact wrench or centroidal momentum. Explicitly incorporating contact point planning into the solution introduces the complementarity relaxation problem, which is still challenging to solve in real time [7]. In some works, the constraints of the LIPM have been relaxed, for example, the introduction of variable-height capturability analysis in [8], which, however, requires a dedicated sequential quadratic programming solver for nonlinear optimization problems. The improved model proposed in [9] extends the LIPM to adapt to low-friction environments and designs a stepping controller for a planar five-link point-foot robot. Additionally, in recent years, the fusion of multi-strategy approaches (stepping, ankle, hip) has become a research hotspot [10,11]. However, these methods typically utilize multiple reduced-order models to represent different inputs (such as ZMP and equivalent ground reaction forces). How to address the dynamic and kinematic consistency between different models still requires further research.

The step-to-step dynamics introduced in [12] provides another perspective for stepping controllers. Its concept aligns with the Poincaré return map [13] in the control community, indicating that the stability of periodic walking is not instantaneous, but is stable from footstrike to footstrike. The Poincaré map maps the state $x_{k}$ at a specific point on the Poincaré section within one step to the state $x_{k+1}$ at the same point in the next step. The periodic orbit creates a fixed point, and the stability of the orbit is equivalent to the stability of the Poincaré map.

The S2S dynamics of the full-order model are high-dimensional and nonlinear. Therefore, Ref. [12] proposed using HLIP for approximation, ensuring the tracking error converges to an error invariant set to achieve orbital stability. To improve the approximation accuracy of the model, linear regression [14] and support vector machines [15] are used to learn a new linear or polynomial S2S dynamics. These methods have improved the tracking accuracy of the controller to some extent; however, when approximating with a single time-invariant linear system, errors always exist, especially when walking speed increases, height changes, and load varies. The question of whether it is possible to design control law that fully consider such uncertainty and provide strict robustness guarantees is the focus of this paper.

Direct data-driven control based on behavioral system theory [16] provides a new possible perspective for this problem. Unlike traditional adaptive control [17], direct data-driven control bypasses the identification step, constructing feedback control laws directly from data while guaranteeing the stability and performance of the unknown system. The fundamental results of Willems et al. [18] on noiseless data of linear systems have led to widespread research in the control community in recent years [19]. Designing robust feedback controllers using noisy input-state data is a current research hotspot in this field. Linear matrix inequalities (LMIs) are used to provide robust guarantees for data-consistent uncertain linear systems, such as the S-lemma [20] and Petersen’s lemma [21]. Improving the dynamic performance of control systems is also being studied, such as adjusting the LMI region [22] and enhancing $H_{2}$ [23] or $H_{\infty }$ performance [20].

Contribution: In this letter, we directly construct a robust state-feedback controller using input-state data of BRUCE’s flat-ground walking collected from simulations with a nominal controller. The deviation between the linear system and the nonlinear system due to model uncertainty can be considered as uncertain noise, which is approximated by a bounded energy ellipsoid. By using robust data-driven control (RDDC) [21], we design a semi-definite programming (SDP) problem to solve for the state-feedback gain that stabilizes a set of data-consistent (A,B). Here, (A,B) represents a family of matrices equivalently characterized by bounded energy ellipsoids, rather than the deterministic (A,B) obtained through traditional parameter identification or the (A,B) used in HLIP. We provide a detailed explanation in the following text for Equation (Equation 5). We integrate the data-driven stepping controller into the whole-body control (WBC) to generate joint torques, and verify the improvement in robustness performance through experiments in different scenarios, such as push recovery, varying loads, and slopes, just as is shown in Figure 1.

## 2. Bipedal Robot Model and Control Framework

### 2.1. Bipedal Robot Model

BRUCE [24] is a small, low-cost humanoid robot consisting of a torso and two 5-DoF legs. Presented in Figure 2, each leg includes a spherical hip joint, a single-DoF knee joint, and a single-DoF ankle joint. Each foot is designed to make line contact with the ground to prevent overconstraint. Each joint is equipped with proprioceptive drive using BEAR actuators [24]. Relevant physical parameter details are listed in Table 1.

### 2.2. Control Framework Overview

BRUCE’s actuator is positioned near the hips, thus reducing the leg inertia. However, as shown in Table 1, the inertia of the legs relative to the torso is still significant, which actually contradicts the assumptions of LIP. This introduces model uncertainties, motivating us to introduce robust data-driven control.

The entire control framework is illustrated in Figure 3. We can use a nominal controller to make BRUCE walking in simulation and collect the dataset $D_{\mathrm{origin}}$ of CoM state and step-length pairs. Gaussian noise with bounded amplitude is added to $D_{\mathrm{origin}}$ to obtain the noisy dataset $D_{\mathrm{noise}}$. Offline, we solve for robust state feedback gains *K*. During online operations, the Stepping Controller is used to solve for step length in real time, and a fifth-order polynomial interpolation is employed to obtain foot trajectories. At the low level, the Whole-Body Controller calculates joint torques to track the trajectories in task space.

## 3. Robust Stepping Controller Design

### 3.1. S2S Dynamics in Data-Driven Perspective

The dynamics of bipedal walking are hybrid, encompassing both continuous dynamics from swing leg lift-off to near landing (pre-impact) and discrete switching dynamics around ground contact. For ROMs like LIP, we consider only the evolution of the center of mass (CoM), where a complete S2S dynamics describes the state transition from the *k*-th step’s pre-impact state to the (k+1)-th step’s pre-impact state. The dynamics in the *x* and *y* directions are decoupled and can be approximated separately as
(1)$$x_{k+1}=Ax_{k}+Bu_{k}$$
where $x_{k}\in R^{2}$ represents the position of the CoM relative to the stance leg contact point and the CoM velocity (or alternatively, the momentum of the CoM relative to the contact point) in the decoupled *x*/*y*-axis. $u_{k}\in R^{2}$ represents the step length.

**Remark** **1.**
*$A\in R^{2\times 2}$ and $B\in R^{2\times 1}$ represent arbitrary linear approximations of S2S dynamics. For the LIPM/ALIP forms, please refer to [12] and [25], respectively.*


In this letter, we provide the deterministic data-driven form of $A^{*}$ and $B^{*}$. Assume the pre-impact state and step length data collected at each step from simulations form the following matrices:(2)$$\begin{array}{c} X_{0}:=\left[x_{d}\left(t_{0}\right),x_{d}\left(t_{1}\right)\ldots x_{d}\left(t_{T-1}\right)\right]\\ U_{0}:=\left[u_{d}\left(t_{1}\right),u_{d}\left(t_{2}\right)\ldots u_{d}\left(t_{T-1}\right)\right]\\ X_{1}:=\left[x_{d}\left(t_{1}\right),x_{d}\left(t_{2}\right)\ldots x_{d}\left(t_{T-1}\right)\right] \end{array}$$

The error of the data-driven S2S dynamics relative to the real system’s S2S dynamics can be regarded as an unknown disturbance sequence, denoted by $W_{0}:=\left[\omega_{t_{0}},\omega_{t_{1}}\ldots ,\omega_{T-1}\right]$. $W_{0}$ satisfies
(3)$$X_{1}=A^{*}X_{0}+B^{*}U_{0}+W_{0}$$

Assume the disturbance sequence $W_{0}$ has bounded energy, i.e., $W_{0}\in W$, where, for some matrix Δ,
(4)$$W:=\left\{W\in R^{n\times T}:WW^{T}⪯\Delta \Delta^{T}\right\}$$

The set C of all matrix pairs consistent with the data (Equation 2) and set W can be represented as
(5)$$C:=\left\{\left[A\;B\right]:X_{1}=AX_{0}+BU_{0}+W,W\in W\right\}$$

We note that $[A^{*}\;B^{*}]\in C$, which is a matrix ellipsoid. At this point, we have obtained a viewpoint that differs from the conventional notion of S2S approximation. Specifically, S2S dynamics are represented by a set of matrices C, rather than a specific *A* and *B*. From the control synthesis perspective, we also need to stabilize the entire family of systems contained in this set.

### 3.2. Target Pre-Impact States in Data-Driven Form

As shown in Equation (Equation 1), the input to the S2S dynamics of the bipedal robot is only the step length. Therefore, it is necessary to adjust the step length to achieve the desired CoM state. From the perspective of periodic stability [12], a stable periodic walking pattern is achieved by solving for the step length to ensure that the robot returns to the current pre-impact state after completing 1 step (or 2 steps) with $x_{k}=x_{k+1}$($x_{k}=x_{k+2}$), thus generating the *P1-Orbit* and *P2-Orbit* [12]. For an intuitive explanation and the physical significance of the two orbits, see Figure 2. In the design of the stepping controller in this paper, the *P1-Orbit* is used to generate forward and backward walking, while the *P2-Orbit* is used to generate lateral walking (for lateral walking, it takes two steps to return to the initial state). Refer to [12] for a detailed explanation of this setting.

**Lemma** **1**([21])**.**
*According to the certainty equivalence principle, the matrix ellipsoid center of (Equation 5) is actually equivalent to the solution of the following least squares problem:*
(6)$$\begin{array}{cc} \left[A_{ls},B_{ls}\right] & :=\arg\min_{\left[A\;B\right]}\left|\right|X_{1}-AX_{0}-BU_{0}{\left|\right|}_{F}^{2}\\ \; & =X_{1}{\left[\begin{array}{c} X_{0}\\ U_{0} \end{array}\right]}^{†}=\left[A^{*},B^{*}\right]+W_{0}{\left[\begin{array}{c} X_{0}\\ U_{0} \end{array}\right]}^{†} \end{array}$$*where*
^†^
*represents the Moore–Penrose pseudoinverse of the matrix. According to the definitions of the P1-orbit and P2-Orbit, we can obtain the target CoM states of the two orbits given the desired step length $u^{*}$:*
(7)$$\begin{array}{cc} \; & x_{p_{1}}^{*}=A_{ls}x_{p_{1}}^{*}+B_{ls}u^{*} \end{array}$$
(8)$$\begin{array}{cc} \; & x_{p_{2}}^{*}=A_{ls}\left(A_{ls}x_{p_{2}}^{*}+B_{ls}u_{l/r}^{*}\right)+B_{ls}u_{r/l}^{*} \end{array}$$

For the desired velocity usr-input $v_{d}$ and swing time $T_{ssp}$, in the case of the *P1-Orbit*, $u^{*}=v_{d}\cdot T_{ssp}$, and for the *P2-Orbit*, $u_{l/r}^{*}+u_{r/l}^{*}=v_{d}\cdot T_{ssp}$. We can solve the above equations to obtain the target CoM states for the two orbits, respectively: $x_{p_{1}}^{*}={\left(I-A_{ls}\right)}^{-1}u^{*}$ and $x_{p_{2}}^{*}={\left(I-A_{ls}^{2}\right)}^{-1}\left(ABu_{l/r}+Bu_{r/l}\right)$.

### 3.3. Robust Data-Driven Feedback Controller Design

In order to ensure that the bipedal robot robustly stabilizes to a periodic orbit, we propose the following stepping controller:(9)$$u_{k}^{R}=u_{k}^{*}+K\left(x_{k}^{R}-x_{k}^{*}\right)$$
where $x_{k}^{*}$ and $u_{k}^{*}$ are the target state and step length at step *k* solved by (Equation 7) in *the P1-Orbit* or (Equation 8) in the *P2-Orbit*, $x_{k}^{R}$ and $u_{k}^{R}$ are the pre-impact state and step length of the current step, and $K\in R^{2}$ is the feedback gain to make the system stable.

**Theorem** **1.**
*When the closed-loop matrix $A^{*}+B^{*}K$ is Schur-stable, the controller (Equation 9) ensures that the system converges to a disturbance invariant set E.*


**Proof.** The system obtained in (Equation 6) can be written as
(10)$$\begin{array}{cc} x_{k+1}^{ls} & =A_{ls}x_{k}^{ls}+B_{ls}u_{k}^{ls}=\left[A_{ls}\;B_{ls}\right]\left[\begin{array}{c} x_{k}^{ls}\\ u_{k}^{ls} \end{array}\right] \end{array}$$
(11)$$\begin{array}{cc} \; & ={\left(\left[A^{*},B^{*}\right]+W_{0}\left[\begin{array}{c} X_{0}\\ U_{0} \end{array}\right]\right)}^{†}\left[\begin{array}{c} x_{k}^{ls}\\ u_{k}^{ls} \end{array}\right] \end{array}$$The uncertainty model defined as in (Equation 5) can be represented as
(12)$$\begin{array}{cc} x_{k+1}^{R} & =A^{*}x_{k}^{R}+B^{*}u_{k}^{R}+w_{k} \end{array}$$Setting $e_{k}=x_{k+1}^{R}-x_{k+1}^{ls}$, substituting (Equation 9) into (Equation 12), and taking the difference with (Equation 11), we obtain the error dynamics:
(13)$$e_{k+1}=\left(A^{*}+B^{*}K\right)e_{k}-W_{0}{\left[\begin{array}{c} X_{0}\\ U_{0} \end{array}\right]}^{†}\left[\begin{array}{c} x_{k}^{ls}\\ u_{k}^{ls} \end{array}\right]+w_{k}$$Since $x_{k}$, $u_{k}$, and $w_{k}$ are bounded in bipedal walking,
(14)$$-W_{0}{\left[\begin{array}{c} X_{0}\\ U_{0} \end{array}\right]}^{†}\left[\begin{array}{c} x_{k}^{ls}\\ u_{k}^{ls} \end{array}\right]+w_{k}\in \Omega$$If $A^{*}+B^{*}K$ is Schur-stable, the error dynamics has a minimum disturbance invariant set *E*:
(15)$$(A^{*}+B^{*}K)E⊕\Omega \in E$$□

**Theorem** **2.**
*For data given by $X_{0}$, $U_{0}$, $X_{1}$ in (Equation 2), if matrix $\left[\begin{array}{c} X_{0}\\ U_{0} \end{array}\right]$ has full rank, the Schur stability of $A^{*}+B^{*}K$ equivalent to the following SDP problem is solvable,*

(16)
$$\begin{array}{cc} find\; & J,\;P=P^{T}≻0\\ s.t.\; & \left[\begin{array}{ccc} -P-D_{11} & 0 & D_{12}\\ 0 & -P & \left[P\;J^{T}\right]\\ D_{21} & \left[\begin{array}{c} P\\ J \end{array}\right] & -D_{22} \end{array}\right]≺0 \end{array}$$

*where we define*

(17)
$$\begin{array}{c} \left[\begin{array}{cc} D_{11} & D_{12}\\ D_{21} & D_{22} \end{array}\right]:=\left[\begin{array}{cc} X_{1}X_{1}^{T}-\Delta \Delta^{T} & -X_{1}{\left[\begin{array}{c} X_{0}\\ U_{0} \end{array}\right]}^{T}\\ -\left[\begin{array}{c} X_{0}\\ U_{0} \end{array}\right]X_{1}^{T} & \left[\begin{array}{c} X_{0}\\ U_{0} \end{array}\right]{\left[\begin{array}{c} X_{0}\\ U_{0} \end{array}\right]}^{T} \end{array}\right] \end{array}$$

*and the feedback gain is $K=JP^{-1}$.*


**Proof.** See Theorem 1 in reference [21]. □

## 4. Trajectory Generation and Whole-Body Control

We can generate the next-step-length input $u_{k}$ at each pre-impact state on both the *x* and *y* axes using (12). The desired swing foot trajectory $x_{f}\in R^{3}$, ${\dot{x}}_{f}\in R^{3}$ can be obtained by the fifth-order polynomial interpolation [24]. The desired CoM trajectory $x_{CoM}\in R^{3}$, ${\dot{x}}_{CoM}\in R^{3}$ and torso angular trajectory $R_{torso}\in R^{3\times 3}$, $\omega_{torso}\in R^{3}$ are specified by the user. The yaw angle of the swing leg is consistent with the torso, and the pitch and roll directions remain constant at zero, resulting in the swing foot angular trajectory $R_{foot}\in R^{3\times 3}$, $\omega_{foot}\in R^{3}$. The reference acceleration in the task space for a linear/angular task is then tracked using a PD control, given by
(18)$$\begin{array}{ccc} \; & {\ddot{x}}_{lin}^{des}=K_{p}\left(x_{i}^{ref}-x_{i}\right)+K_{d}\left({\dot{x}}_{i}^{ref}-{\dot{x}}_{i}\right) & \end{array}$$
(19)$$\begin{array}{cc} \; & {\ddot{x}}_{ang}^{des}=K_{p}log\left(R_{i}^{T}R_{i}^{ref}\right)+K_{d}\left(\omega_{i}^{ref}-\omega_{i}\right) \end{array}$$
where the $x_{□}^{ref}$ and $x_{□}$ corresponds to the *i*-th task reference trajectories and state estimates. $K_{p}$ and $K_{d}$∈$R^{3\times 3}$ represent the PD gain matrix for different tasks.

The generated task-space trajectories are tracked using whole-body control. The specific QP formulation is as follows:
(20a)$$\begin{array}{cc} \min_{\ddot{q},f_{j}}\; & \sum_{i=1}^{N_{t}}\left|\right|J_{i}\ddot{q}+{\dot{J}}_{i}\dot{q}-{\ddot{x}}_{i}^{des}{\left|\right|}_{W_{i}}^{2}+\sum_{j=1}^{N_{c}}\left|\right|f_{j}{\left|\right|}_{W_{f}}^{2} \end{array}$$
(20b)$$\begin{array}{cc} s.t.\; & S_{f}\left(H\ddot{q}+C\dot{q}+G-\sum_{j=1}^{N_{c}}J_{c_{j}}^{T}f_{j}\right)=0 \end{array}$$
(20c)$$\begin{array}{cc} \; & f_{j}\in U \end{array}$$
(20d)$$\begin{array}{cc} \; & |\ddot{q}|\;\in \;{\ddot{q}}_{max} \end{array}$$
where $J_{i}$ is the *i*-th task Jacobian and $N_{t}$ is the number of tasks, concluding the swing position/orientation task, torso orientation task, linear/angular moment, task and stance contact task. $S_{f}\in R^{6\times 16}$ is the selection matrix associated with the underactuated DoF. The weight of the *i*-th task is set to $W_{i}\in R^{3\times 3}$. The decision variables $\ddot{q}\in R^{16}$ and $f_{j}\in R^{6}$ are the joint acceleration and contact wrench. Specifically, for the stance contact constraint, $J_{c_{j}}\ddot{q}+{\dot{J}}_{c_{j}}\dot{q}=0$ is also set as a term in the cost function in the form of a soft constraint. The rate of change of the centroidal angular momentum is defined as ${\dot{k}}^{des}=-K_{d}k$ to suppress excessive angular momentum, and the centroidal momentum matrix [5] is used as the task Jacobian.

$f_{j}$ is the 6D force wrench for the *j*-th contact foot. For BRUCE’s line-foot, we use the contact wrench cone (CWC) described in [26] as the constraint in ([Disp-formula FD20c-biomimetics-09-00681]).

By solving the QP problem, we obtain the optimal solutions ${\ddot{q}}^{*}$ and $f_{j}^{*}$, which can be used to calculate the joint torques through the joint dynamics of the system:(21)$$\begin{array}{c} \tau^{*}=S_{a}\left(H{\ddot{q}}^{*}+C\dot{q}+G-\sum_{j=1}^{N_{c}}J_{c_{j}}^{T}f_{j}^{*}\right) \end{array}$$
where $S_{a}\in R^{10\times 16}$ is the selection matrix associated with the actuated DoF.

## 5. Results

### 5.1. Data Collection, Parameter Selection, and Feedback Gain

The collection of Hankel matrices in Equation (Equation 2) was conducted in the Gazebo simulator using a standard HLIP-based stepping controller [12], performing level walking in decoupled lateral and sagittal directions. For each direction, $T_{sample}=1500$ steps were collected to ensure persistence of excitation [18], with random assignments of desired velocity controlled via keyboard input within a speed range of [−0.15,0.15] m/s.

The robustness of the stepping controller depends on the coverage range of the set C in (Equation 5). For the dataset collected from level walking, where the bounded disturbances of both linear and nonlinear systems are unknown, we roughly set a range ${\left[d_{1/2}^{min},d_{1/2}^{max}\right]}^{T}$ corresponding to each component of Δ in Equation (Equation 4).

To enhance the stability of the feedback controller over a broader range of $\left[A^{*},B^{*}\right]$, we introduced bounded Gaussian noise ${\left[\delta \omega_{1},\delta \omega_{2}\right]}^{T}∼{\left[N\left(\mu_{1},\sigma_{1}^{2}\right),N\left(\mu_{2},\sigma_{2}^{2}\right)\right]}^{T}$ into the original state data $X_{1}$, thereby increasing uncertainty, where the $\delta \omega_{1/2}\in \left[\delta \omega_{1/2}^{min},\delta \omega_{1/2}^{max}\right]$. This proved effective in practical simulations. Then, we noticed that each component of Δ should increase the upper bound of this Gaussian noise. In practical systems, excessively large noise bounds can render SDP infeasible; thus, we ultimately set the parameters as $\left[d_{1}^{min},d_{1}^{max}\right]=\left[-0.002,0.002\right]$, $\left[d_{2}^{min},d_{2}^{max}\right]=\left[-0.01,0.01\right]$, $\left[\delta \omega_{1}^{min},\delta \omega_{1}^{max}\right]=\left[-0.004,0.004\right]$ and $\left[\delta \omega_{1}^{min},\delta \omega_{1}^{max}\right]=\left[-0.006,0.006\right]$. Therefore, each component of $\tilde{\Delta }$ with the introduced noise is equal to the matrix formed by the addition of the corresponding components of both.

**Remark** **2.**
*We use the original dataset $D_{origin}$ without introducing additional noise to calculate the target state in order to prevent the invariant disturbance set from expanding. The dataset $D_{noise}$ with introduced noise, along with $\tilde{\Delta }{\tilde{\Delta }}^{T}$, is used to calculate the gain K, stabilizing the system under larger C to counteract additional disturbances due to model uncertainties.*


The robust feedback gain *K* is then determined through offline solving (Equation 16) using CVXPY [27]. The WBC in ([Disp-formula FD20d-biomimetics-09-00681]) is solved in real time with 500 Hz using OSQP [28]. Notably, the SDP problem is solved offline, and the obtained feedback gain *K* is used to synthesize a deadbeat-style online stepping controller (9), ensuring real-time performance.

**Remark** **3.**
*Theoretically, the larger the added noise, the greater the maximum uncertainty that the stepping controller can stabilize, provided that Problem (16) remains feasible. However, excessive noise can render Problem (16) infeasible. Therefore, in our experiments, we selected the maximum noise that ensures the feasibility of Problem (16) as a parameter. Since the data collected from the simulator can be approximated as noise-free, the magnitude of the added bounded energy ellipsoidal noise serves as the boundary for the maximum disturbance that the system can encounter during experiments.*


### 5.2. Simulation Results

#### 5.2.1. Heavy Payload Task

The most direct manifestation of model uncertainty lies in the adaptability to different mass and inertia. From this perspective, the addition of noise in data collection is similar to the neighborhood adaptation concept in the field of reinforcement learning [29]. We hope that the stepping controller can adapt to a specific amount of payload without the need to re-collect a large amount of data for that specific payload. This situation can be considered as an uncertainty in the linear system $\left[A^{*},B^{*}\right]$, which is uniformly stabilized through our method.

Figure 4 illustrates the adaptability of different algorithms on BRUCE when increasing its mass to alter its overall inertia. For comparison, data-driven control under the assumption of no noise [16] is also considered. To ensure statistical reliability, each algorithm runs 30 times in different scenarios. When the mass variation is small, HLIP and noise-free DDC can both ensure a certain success rate in 20 m walking. However, as the mass increases beyond 1 kg (approximately one-fifth of the entire mass of BRUCE), the robustness of our method allows the controller to overcome disturbances caused by changes in model parameters.

#### 5.2.2. Sloping Terrain Task

Walking on slopes introduces acceleration ${\ddot{z}}_{CoM}$ in BRUCE’s vertical direction, thereby altering the S2S dynamics. We did not make any special adjustments to the swing leg trajectory generation, causing early touchdown when the desired speed increases.

Figure 5 shows the maximum walking speeds of the RDDC and HLIP stepping controllers under different slopes. As velocity increases, vertical acceleration in the *z*-direction also increases, leading to a shorter swing time $T_{ssp}$. The simulation results demonstrate that RDDC exhibits better robustness under the influence of two types of uncertainty parameters.

#### 5.2.3. Push Recovery Task

In this section, we compare the lateral and lateral disturbance rejection capabilities of the HLIP/RDDC-based stepping controller. In simulation experiments, BRUCE tracked the same variable velocity command. The CoM offsets of our method and HLIP initial controller under different directions and external forces are shown in the Figure 6 and Figure 7. External forces were applied at the beginning of each step cycle for the same duration $t_{\mathrm{force}}=20\;\mathrm{ms}$, approximately one-tenth of a step cycle, simulating the impact of external forces on the actual system.

**Remark** **4.**
*The HLIP-based method [12] can achieve locomotion capabilities similar to RDDC during BRUCE’s walking on flat ground. However, due to the larger ratio of leg inertia to total body inertia in BRUCE compared to Cassie [12], it is more difficult to use LIP-like models to capture key physical characteristics. The DDC method, to some extent, achieves a more accurate linear system. At the same time, the robust approach based on ellipsoidal noise captures the effects of more general model uncertainties. Therefore, in experiments across various environments, our method outperforms the HLIP method.*


#### 5.2.4. Comparison with Indirect Data-Driven Methods

Unlike direct data-driven methods, indirect data-driven control [14] typically involves two steps: offline parameter identification to obtain the system dynamics parameters, followed by computation of the feedback gain K. The intuitive procedural differences between the two approaches are illustrated in Figure 8 below.

#### 5.2.5. Sensitivity Analysis Under Different Noise Levels

We can solve for the stable control gain *K* under the given range of uncertain noise (Equation 4) using Equation (Equation 16). Its stability can be naturally obtained through the Lyapunov stability condition for discrete systems [21].

To verify the maximum noise level that ellipsoids of different noise levels can withstand, we added various levels of noise to the collected original data. Correspondingly, the size of the bounded energy ellipsoid also changed. We compared the maximum noise levels that can be resisted under the HLIP model for different gains *K* derived by solving the SDP problem constructed with bounded energy ellipsoids after adding different noise levels. The corresponding results are shown in Figure 9 below.

By directly testing on the HLIP, we can clearly observe the response changes under different disturbances, demonstrating the sensitivity of bounded energy ellipsoids of varying sizes to perturbations. We observed that adjusting the size of the bounded energy ellipsoid helps to withstand larger disturbances. Although the improvement is modest, it has been shown to enhance the robustness of the walking controller in the actual BRUCE simulation.

#### 5.2.6. Sim-to-Sim Experiments

We also conducted sim-to-sim experiments from Gazebo to Mujoco, calculating feedback gains from data collected under different lengths. First, walking was performed in Gazebo using a nominal controller (HLIP), and data were collected for 500 steps (−0.1 m/s to −0.1 m/s), 1000 steps (−0.2 m/s to −0.2 m/s), and 3000 steps (−0.2 m/s to −0.2 m/s). We observed that stable walking in Mujoco can only be achieved when a larger amount of data is collected and broader range of walking speeds is covered.

The feedback gains obtained by solving (16) differs based on the length of the collected data, as shown in Table 2 below; refer to Figure 10. We increased the complexity of the walking scenario in Mujoco by adding a 0.02 m step to the ground. When the amount of collected data is small, the controller causes a fall after walking for some time. However, with a larger data sample, BRUCE can successfully traverse this terrain.

## 6. Conclusions and Future Work

In this paper, we directly construct a state-feedback controller using input-state data from S2S dynamics during bipedal walking, along with an additional noise sequence. Given the range of uncertainties, we can design robust feedback gains to stabilize a family of linear systems. Our work not only demonstrates the potential of behavior system theory in gait control for legged robots, but also shows that this data-driven control method, compared to other related data-driven approaches such as deep reinforcement learning or supervised learning, does not require large-scale data and training costs, achieving robustness similar to or even better than model-based control methods. Moreover, we have developed a mathematically rigorous robust controller that theoretically characterizes the possible disturbances arising from model uncertainties. Unlike machine learning’s “black box”, this method provides a more comprehensive explanation for specific types of model uncertainties. At the same time, compared to robust model predictive control, this method does not require online solutions to robust optimal control problems (such as min–max optimal control), thereby improving computational real-time performance.

Through several representative simulation scenarios, the potential of behavior system theory-based data-driven control in humanoid locomotion control has been demonstrated.

However, the current method still has some limitations and challenges, which we will address in future work.

We found that the stepping controller based on RDDC does not show significant improvement in speed tracking compared to the HLIP-based controller. This is because we only used the ellipsoidal matrix center of *C* as the reference dynamics. When the uncertainty range of *C* is large, it results in a large error invariant set.The S2S dynamics are inherently nonlinear. If a nonlinear mathematical structure is determined, a state-feedback controller can be constructed by feedback linearization of this system using [30].The system behavior obtained by the nominal controller (HLIP or other stepping controllers) is limited. When walking at higher speeds, the nonlinear effect of the swing leg inertia increases. DeePO [31] can be used for online updating of the *K* value to address this issue.

## Figures and Tables

**Figure 1 biomimetics-09-00681-f001:**
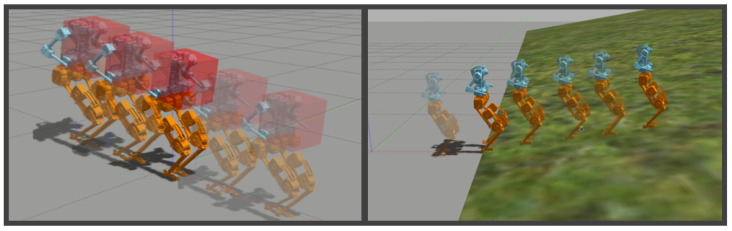
BRUCE carrying payload and walking up a slope.

**Figure 2 biomimetics-09-00681-f002:**
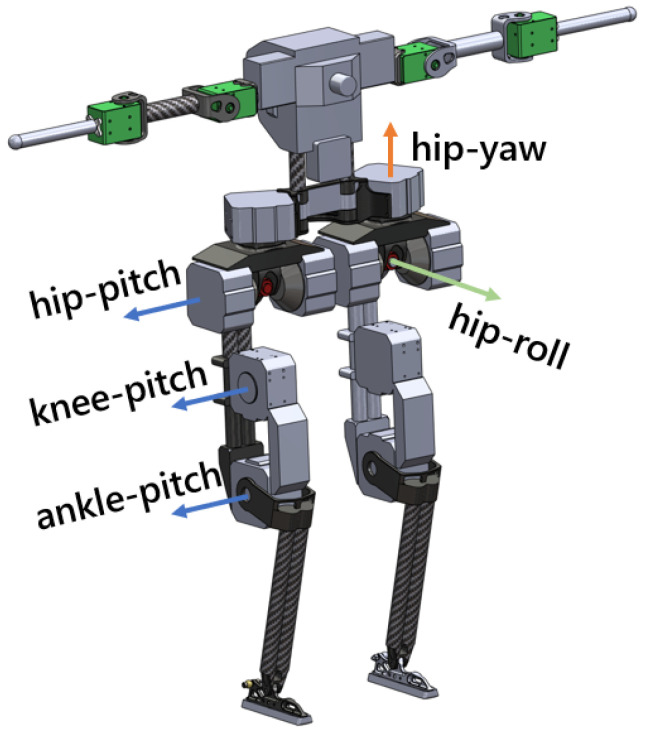
Configuration of BRUCE robot.

**Figure 3 biomimetics-09-00681-f003:**
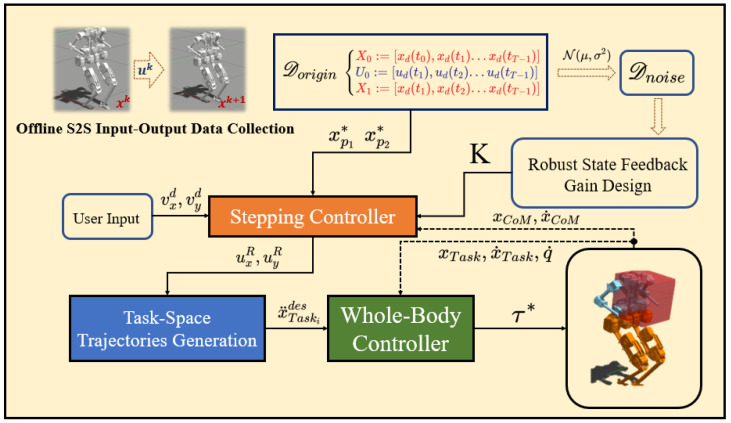
The proposed control framework.

**Figure 4 biomimetics-09-00681-f004:**
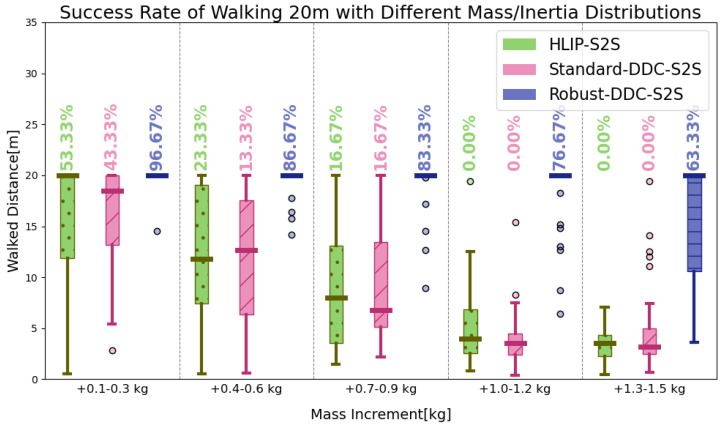
The success rate of walking forward for 20 m with different stepping controllers.

**Figure 5 biomimetics-09-00681-f005:**
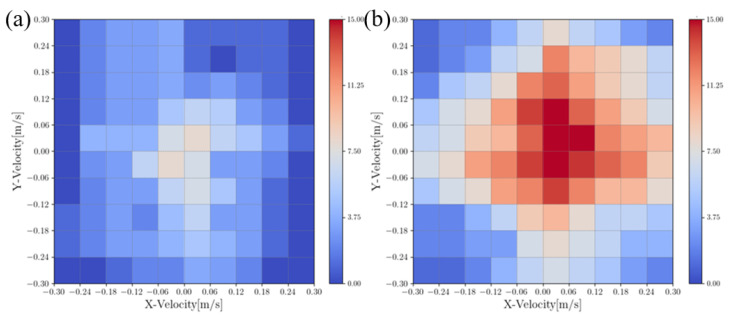
A heat map of maximum walking speeds under varying slopes: (**a**) HLIP stepping controller; (**b**) RDDC stepping controller.

**Figure 6 biomimetics-09-00681-f006:**
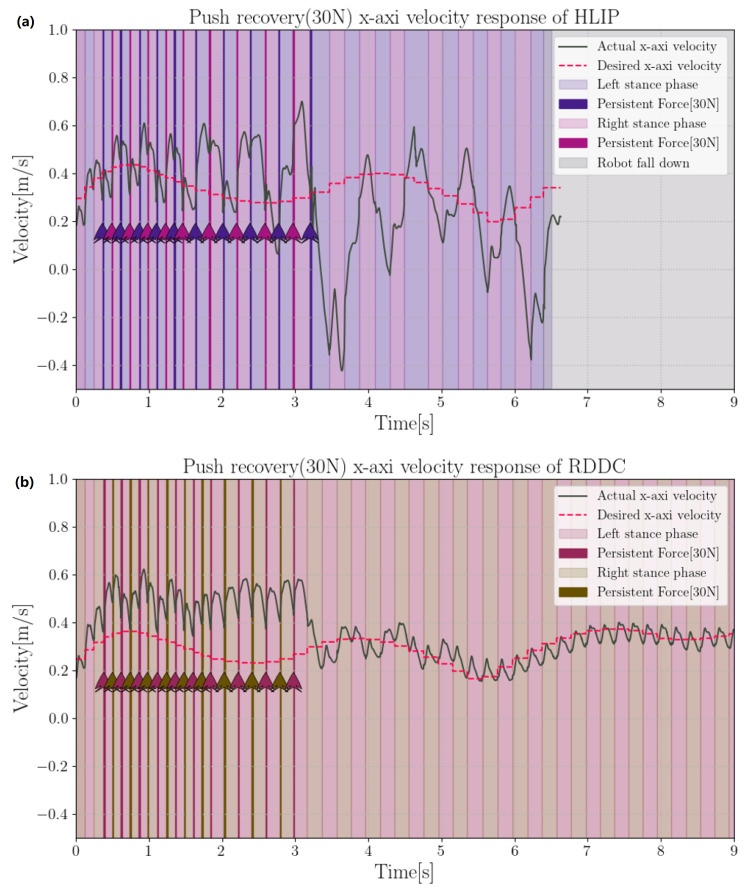
Push recovery of 30 N in *X*-direction: (**a**) HLIP stepping controller; (**b**) RDDC stepping controller.

**Figure 7 biomimetics-09-00681-f007:**
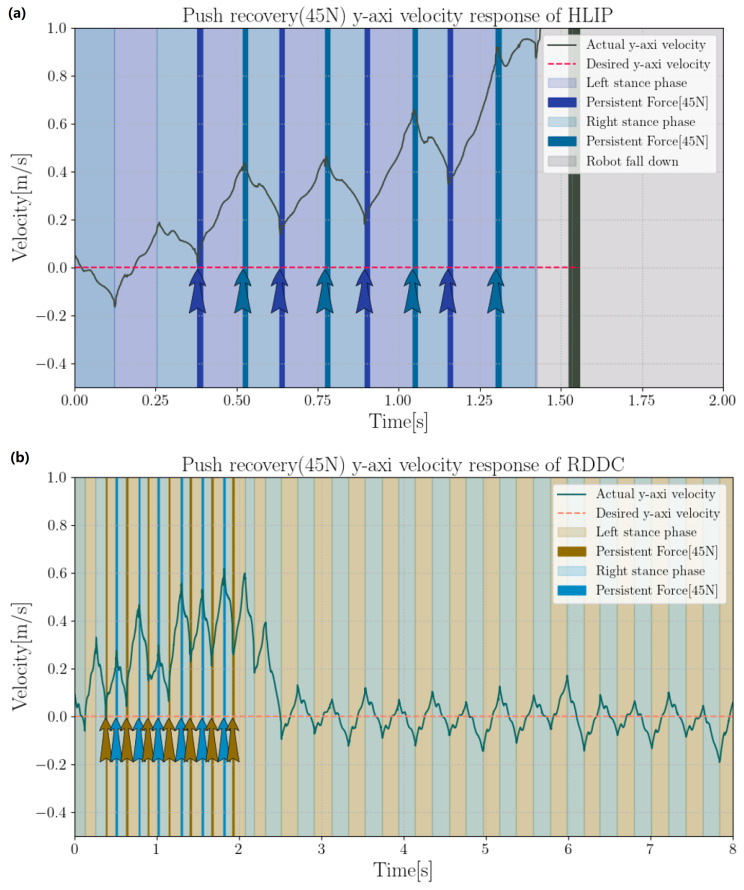
Push recovery of 45 N in *Y*-direction: (**a**) HLIP stepping controller; (**b**) RDDC stepping controller.

**Figure 8 biomimetics-09-00681-f008:**
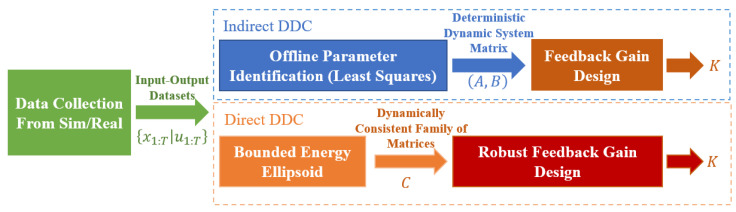
The differences between the direct and indirect data-driven control frameworks are as follows. In direct data-driven control, the matrix family C is used only as an intermediate parameter, represented by bounded energy ellipsoidal parameterization, and is not solved explicitly. This represents a fundamental difference from indirect data-driven control.

**Figure 9 biomimetics-09-00681-f009:**
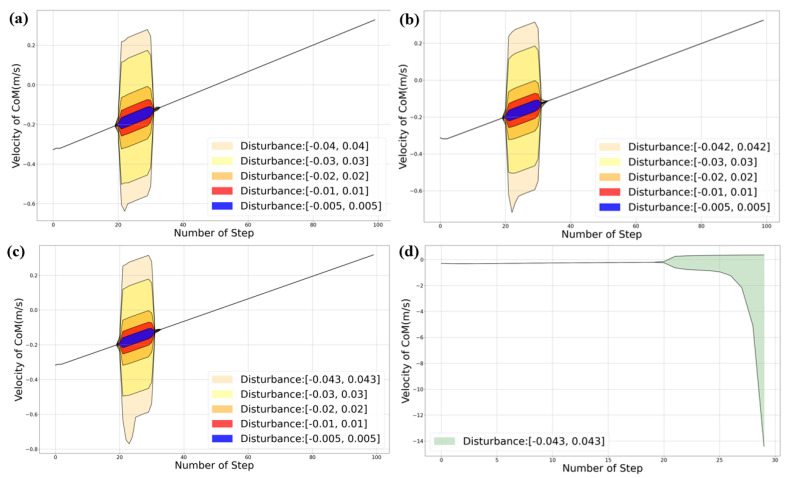
The results in figures (**a**–**c**) show the velocity responses of the HLIP after walking within the same speed range, subjected to disturbances with added noise ranges with absolute values 0.04, 0.042, and 0.043, respectively. The ellipsoid energy level matches the noise range in each case. Figure (**d**) illustrates that when the noise range is 0.04, the system diverges under disturbances of the corresponding magnitude.

**Figure 10 biomimetics-09-00681-f010:**
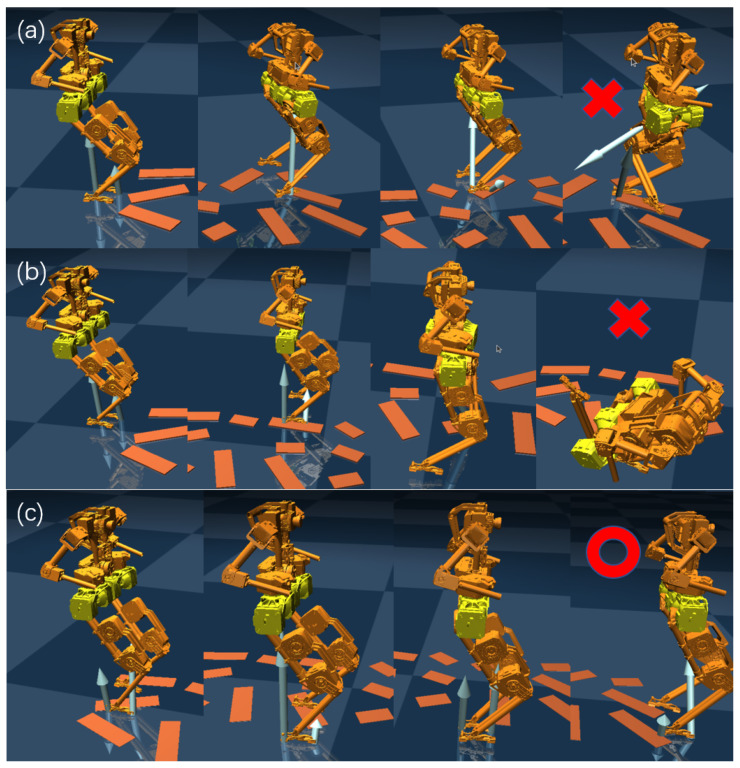
Sim-to-sim experiments in Mujoco using a stepping controller built from Gazebo data of varying lengths: (**a**) 500 steps, (**b**) 1000 steps, (**c**) 3000 steps.The red circle and cross marks represent successful passes and failures due to falls, respectively.

**Table 1 biomimetics-09-00681-t001:** Physical parameters of robot.

Parameter	Symbol	Value
Torso Mass (kg)	$M_{torso}$	1.31
Each Leg Mass (kg)	$M_{leg}$	1.44
Torso Inertia (kg·m^2^)	$I_{xx}$/$I_{yy}$/$I_{zz}$	0.013/0.010/0.005
Leg Inertia (kg·m^2^)	$I_{xx}$/$I_{yy}$/$I_{zz}$	0.021/0.020/0.001
Rate of Mass	$M_{leg}$/$M_{torso}$	1.09

**Table 2 biomimetics-09-00681-t002:** Feedback gain of different lengths of collected data.

Number of Steps	Feedback Gain	Success or Failure
500	[1.075,0.43]	Failure
1000	[1.205,0.38]	Failure
3000	[1.375,0.26]	Success

## Data Availability

The data and simulation code used to support the findings of this study are available from Chao Song upon request.

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
