# Peer review of "Bipedal Stepping Controller Design Considering Model Uncertainty: A Data-Driven Perspective"

_biomimetics, 2024, doi:10.3390/biomimetics9110681_

Round 1
Reviewer 1 Report
Comments and Suggestions for Authors
This work utilized the robust data-driven control method to mitigate the model mismatch between the simplified model and the whole-body model. Then, a novel step-to-step dynamics control method is proposed to realize robust locomotion.
Overall, the work is well-organized, and the motivation is clearly stated. However, the methodology should be clarified better, and more experiments should be added. Major problems should be addressed.
(1) More advanced models should be discussed in the introduction section. For example, the variable-height inverted pendulum (Ref. 1) and the nonlinear inverted pendulum pulse flywheel model (Ref. 2) have been widely used.
Ref. 1: Caron, S., Escande, A., Lanari, L. and Mallein, B., 2019. Capturability-based pattern generation for walking with variable height. IEEE Transactions on Robotics, 36(2), pp.517-536.Ref. 2: Ding, J., Han, L., Ge, L., Liu, Y. and Pang, J., 2022. Robust locomotion exploiting multiple balance strategies: An observer-based cascaded model predictive control approach. IEEE/ASME Transactions on Mechatronics, 27(4), pp.2089-2097.
(2) The Methodology should be better formulated.
-- In (6), what does "{}^{+}" mean? Matrix transpose or inverse?
-- In this work, is the P2-orbit dynamics in (8) used?
-- In Section 4, you mentioned: "The desired CoM trajectory xCoM ∈ R3, xË™CoM ∈ R3 and torso angular trajectory Rtorso ∈ R3×3, ωtorso ∈ R3 are specified by the user ". How?
--- In Section 4, it seems that the whole-body controller considers multiple tasks, such as swing position/orientation task, torso orientation task, linear/angular moment task and stance contact task. However, you did not mention how to balance these goals. Can you discuss the priority assignment in more detail?
(3) The results are not fully validated
--Since no hardware experiments have been added, I am not convinced of the advantages. Can this data-driven method really capture the locomotion dynamics? I strongly suggest adding hardware tests in the future version. Or Can you directly transfer the motion in the gazebo to another simulator, e.g., Mujuco, without collecting the data from scratch again?
--- In data collection (Section 5.1), did you try periodic motion with different speeds or directions? Does it influence the accuracy of the modelling?
-- Please present the comparison studies among the HLIP-s2s, standard-DDC-s2s and Robust-DDC-s2s in Section 5.2.2 and 5.2.3, as you did in Section 5.2.1.
-- In Figure 6(b), it seems that the robot walks at a higher frequency before 2s. However, you did not mention how to change the stepping frequency before. Please clarify this.
(4) Please check the equations and expressions carefully.
In linear 72, it is unclear what (A, B) means.
In (2), the X_1 should contain x_d(t_T).
ALIP in line 100 should be clearly explained.
In lines 136 and 137, "solved by (7) in x-axis or (8) in y-axis" is not right.
In theorem 2, "....is equivalent to the following SDP problem is solvable" is unclear.
Comments on the Quality of English Language
The English is overall good. Only minor editing of the English language is required.
Author Response
Comments 1 : More advanced models should be discussed in the introduction section. For example, the variable-height inverted pendulum (Ref. 1) and the nonlinear inverted pendulum pulse flywheel model (Ref. 2) have been widely used.
Ref. 1: Caron, S., Escande, A., Lanari, L. and Mallein, B., 2019. Capturability-based pattern generation for walking with variable height. IEEE Transactions on Robotics, 36(2), pp.517-536.
Ref. 2: Ding, J., Han, L., Ge, L., Liu, Y. and Pang, J., 2022. Robust locomotion exploiting multiple balance strategies: An observer-based cascaded model predictive control approach. IEEE/ASME Transactions on Mechatronics, 27(4), pp.2089-2097.
Response 1: The two papers have been added to the article. They are indeed profound works in their respective fields.
(2) The Methodology should be better formulated.
-- In (6), what does "{}^{+}" mean? Matrix transpose or inverse?
-- In this work, is the P2-orbit dynamics in (8) used?
-- In Section 4, you mentioned: "The desired CoM trajectory xCoM ∈ R3, xË™CoM ∈ R3 and torso angular trajectory Rtorso ∈ R3×3, ωtorso ∈ R3 are specified by the user ". How?
--- In Section 4, it seems that the whole-body controller considers multiple tasks, such as swing position/orientation task, torso orientation task, linear/angular moment task and stance contact task. However, you did not mention how to balance these goals. Can you discuss the priority assignment in more detail?
Response 2:
2.1. In (6), the "{}^{+}" means the matrix pseudo-inverse. I have already added an explanation in the original text.
2.2. In the current work, the P1-orbit is used to generate forward and backward walking trajectories, while the P2-orbit is used to generate lateral walking trajectories. Both are applied in the current gait generation algorithm.
2.3. Since the model currently used is the LIPM, with the assumption of constant CoM height, in S2S-Dynamics, the input consists only of step lengths in two decoupled directions. The swing leg trajectory is generated using polynomial trajectories, which is well-defined. In S2S-Dynamics, the desired step length is actually specified by the average velocity v_d can be regarded as the user-specified CoM velocity trajectory. By integrating over time, the CoM position trajectory can be obtained. As for the direction and angular velocity of the torso, these are not reflected in the LIP since it is a point-mass model. Therefore, they need to be separately designed in Whole-Body Control. Generally, it is assumed that there is no change in the roll and pitch directions, so the angular velocity is zero. The yaw angular velocity can be determined according to the robot's orientation during walking. It is assumed that the yaw angular velocity is uniform, and the rotation matrix ​ can be obtained based on the desired final orientation.
2.4. In Whole-Body Control (WBC), the priority is implicitly designed by assigning different weights. First, the contact stability task of the stance leg, usually represented as a hard constraint (as mentioned in the original text), is designed as a relaxed soft constraint in the current WBC for numerical stability. Therefore, this task is assigned the highest weight. For LIP-based trajectory optimization, since the step length is determined by the high-level planner, the swing leg task weight should also be designed relatively high to meet the desired foot placement. For angular momentum and linear momentum tasks, the height along the z-axis should approximately remain constant, and thus the weight for the z-axis is relatively higher than that for the x-y directions. The linear momentum in the x-y directions is specified by the desired CoM velocity and is not actually obtained through planning. If the weight is set too high, it may lead to dynamic inconsistencies, causing walking instability. Overall, the weight design in the implicitly layered WBC still relies on experience, but the general approach aligns with physical principles."
(3) The results are not fully validated
--Since no hardware experiments have been added, I am not convinced of the advantages. Can this data-driven method really capture the locomotion dynamics? I strongly suggest adding hardware tests in the future version. Or Can you directly transfer the motion in the gazebo to another simulator, e.g., Mujuco, without collecting the data from scratch again?
--- In data collection (Section 5.1), did you try periodic motion with different speeds or directions? Does it influence the accuracy of the modelling?
-- Please present the comparison studies among the HLIP-s2s, standard-DDC-s2s and Robust-DDC-s2s in Section 5.2.2 and 5.2.3, as you did in Section 5.2.1.
-- In Figure 6(b), it seems that the robot walks at a higher frequency before 2s. However, you did not mention how to change the stepping frequency before. Please clarify this.
Response 3:
3.1. The debugging of hardware experiments will be conducted in future work. The current objective is to validate the feasibility of using behavioral system theory to capture the S2S-Dynamics of a humanoid robot. I have already supplemented the Mujoco experiments (see the revised version of the paper, Section 5.2.4). The data from Gazebo can be directly used to build the data-driven stepping controller in Mujoco. Hardware experiments will be carried out in future work, with a potential challenge being the discrepancy between the data distribution in simulation (Sim) and reality (Real), which is similar to the problem encountered in reinforcement learning. Moreover, although behavioral system theory is essentially a reformulation of linear systems (and lacks the nonlinear representation capability of neural networks in RL), this control theory is more rigorous. At the same time, this precise description holds the potential to be extended to nonlinear systems. We hope to build upon this work and extend it further in the future.\
3.2. We collected data across the widest possible range of walking speeds for the basic controller. When the speed range of the collected data is too narrow, the final controller may become unstable. This is because we are currently approximating the nonlinear S2S dynamics with a linear model. Higher walking speeds make the influence of the swing leg inertia more pronounced, degrading the performance of the LIP approximation. The data-driven controller built using behavioral system theory can be seen as an average approximation of this nonlinearity across different ranges, making sufficient data samples crucial. Additionally, collecting large amounts of data in simulation is not difficult. For directional movement, we only collected data for forward, backward, and lateral movements, covering four directions in total. Since S2S-Dynamics assumes walking can be decoupled into two directions of motion, there is no need to collect data for more directions
3.3. In Figure 6.b, the time window is larger, which makes the frequency appear lower. In reality, the step frequencies in both cases are identical.
(4) Please check the equations and expressions carefully.
In linear 72, it is unclear what (A, B) means.
In (2), the X_1 should contain x_d(t_T).
ALIP in line 100 should be clearly explained.
In lines 136 and 137, "solved by (7) in x-axis or (8) in y-axis" is not right.
In theorem 2, "....is equivalent to the following SDP problem is solvable" is unclear.
Response 4: These issues have been addressed in the revised manuscript, with annotations added. I appreciate the reviewers' valuable feedback.
4.1. In linear 72, (A, B) means the linear system dynamics matrix of both x/y direction S2S. From a data-driven perspective, A and B are uncertain, and thus they represent a set of data-consistent matrix pairs that satisfy the uncertainty constraints.
Reviewer 2 Report
Comments and Suggestions for Authors
This paper proposes a data-driven gait controller design method for bipedal robots to consider model uncertainty. Researchers directly constructed a robust gait controller by collecting input state data using a nominal controller in a simulation environment. This controller enhances its ability to handle uncertain loads, various slope terrains, and push recovery by representing model uncertainty differences as bounded noise and using a bounded energy ellipsoid for over approximation.
1. The paper successfully applies the theory of behavioral systems to the gait control problem of bipedal robots and proposes a novel data-driven control method. However, the article lacks sufficient discussion on how to extend this method to more complex dynamic environments and real-time applications.
2. The data-driven control strategy proposed in the paper is innovative, but further clarification is needed on its advantages and potential limitations compared to traditional model predictive control methods.
3. The paper's approach to handling model uncertainty is reasonable, but more details are needed to demonstrate the rationality of the selected parameter settings, such as the selection of noise level and energy ellipsoid size.
4. The illustrations in the paper clearly demonstrate the configuration and control framework of the BRUCE robot, but it is recommended to add more illustrations on gait generation and omnidirectional body control to provide readers with a more intuitive understanding of the algorithm implementation.
5. The paper tested different load, slope, and propulsion recovery scenarios in a simulation environment, but lacked comparative experiments with existing methods. It is suggested to increase comparative analysis with existing gait controllers.
6. The paper provides the results of simulation experiments, but does not delve into the factors that affect controller performance, such as the specific impact of gait adjustment strategies and feedback gain selection on performance.
7. In terms of innovative description of algorithms, the article needs to more clearly point out the advantages and innovative points compared to existing methods
8. In "Introduction" section Related Works, I feel the current coverage of the state of the Bipedal Stepping is not satisfactory as the related work section does not cover many contributions that likely provide the building blocks of the proposed approach.
For example,
(1) Balance gait controller for a bipedal robotic walker with foot slipï¼›
(2) LCDL: Towards Dynamic Localization for Autonomous Landing of Unmanned Aerial Vehicle Based on LiDAR-Camera Fusionï¼›
(3) Modeling and Experiments of Bipedal Actuated Linear Piezoelectric Platform With Smooth Motion and Strong Load Capacity.
It is suggested to cite the above articles and analyze the differences in Section Related Works.
9.There are many grammar errors. Please check English through the paper carefully.
Comments on the Quality of English Language
Minor editing of English language required.
Author Response
- The paper successfully applies the theory of behavioral systems to the gait control problem of bipedal robots and proposes a novel data-driven control method. However, the article lacks sufficient discussion on how to extend this method to more complex dynamic environments and real-time applications.
Response 1: The current work primarily focuses on constructing an S2S-style Stepping Controller through data, which essentially involves learning a more robust LIP controller. In humanoid robot motion control, movement in more complex environments often requires more intricate dynamic system models, such as center of mass dynamics or whole-body dynamics, which implies a greater number of input-output variables and, consequently, more data. In the future, the extension of behavioral system theory to nonlinear systems is expected to enable similar work.
In terms of real-time performance, the current stepping controller solves the SDP problem offline, while the online deadbeat controller can meet the real-time requirements.
- The data-driven control strategy proposed in the paper is innovative, but further clarification is needed on its advantages and potential limitations compared to traditional model predictive control methods.
Response 2: Model Predictive Control (MPC) typically requires an explicit dynamic system model to design an optimal control problem that yields the control rate. In contrast, data-driven control based on behavioral system theory constructs a control rate directly from data. Compared to model-based controllers, the advantage of using data lies in its ability to capture the effects of model uncertainties while ensuring robust stability for all data-consistent dynamic systems. This concept is also present in robust MPC, but the control methods from behavioral system theory offer a more direct approach by allowing the construction of stable controllers solely from data. In robust MPC, it usually requires min-max methods to solve more complex optimization problems.
Furthermore, control methods based on behavioral system theory also include MPC approaches, such as DeePC. We plan to utilize DeePC to implement more complex controllers in the future. While DeePC has similar computational complexity to traditional MPC, our current work achieves good walking performance using only state feedback control without the need to solve the optimal control problem online. However, in the future, as we need to address constraint issues, exploring DeePC will remain a worthwhile direction.
- The paper's approach to handling model uncertainty is reasonable, but more details are needed to demonstrate the rationality of the selected parameter settings, such as the selection of noise level and energy ellipsoid size.
Response 3: more rigorous discussion on parameter selection is indeed necessary. I have supplemented this in the article for the reviewers' consideration.
- The illustrations in the paper clearly demonstrate the configuration and control framework of the BRUCE robot, but it is recommended to add more illustrations on gait generation and omnidirectional body control to provide readers with a more intuitive understanding of the algorithm implementation.
Response 4:
The relevant figures and tables have been added to the original text.
- The paper tested different load, slope, and propulsion recovery scenarios in a simulation environment, but lacked comparative experiments with existing methods. It is suggested to increase comparative analysis with existing gait controllers.
Response 5:
The HLIP controller serves as our baseline method for comparison, as it is the most representative approach within the S2S controller framework. I appreciate the reviewers' suggestions, and based on this, I have added comparative experiments involving the ALIP and DCM stepping controllers.
- The paper provides the results of simulation experiments, but does not delve into the factors that affect controller performance, such as the specific impact of gait adjustment strategies and feedback gain selection on performance.
Response 6: Generally speaking, the parameters related to gait adjustment include the stepping period and step length, where the step length serves as the decision variable of the method, while the stepping time is set as a constant within the current framework. The choice of stepping time is usually related to the size of the humanoid robot; for the BRUCE small robot, values between 0.18 and 0.30 seconds are feasible. Regarding the selection of feedback gains, the SDP problem can directly provide the gains that stabilize the current system. However, in practical engineering applications, terms related to speed should not be set too high, as this may lead to oscillations in gait due to overshooting caused by speed deviations
- In terms of innovative description of algorithms, the article needs to more clearly point out the advantages and innovative points compared to existing methods
Response 7: Thank you for the reviewers' suggestions. I have made revisions to the abstract and conclusion in the original text.
- In "Introduction" section Related Works, I feel the current coverage of the state of the Bipedal Stepping is not satisfactory as the related work section does not cover many contributions that likely provide the building blocks of the proposed approach.
For example,
(1) Balance gait controller for a bipedal robotic walker with foot slipï¼›
(2) LCDL: Towards Dynamic Localization for Autonomous Landing of Unmanned Aerial Vehicle Based on LiDAR-Camera Fusionï¼›
(3) Modeling and Experiments of Bipedal Actuated Linear Piezoelectric Platform With Smooth Motion and Strong Load Capacity.
It is suggested to cite the above articles and analyze the differences in Section Related Works.
Response 8: Modifications have been made to the original text.
9.There are many grammar errors. Please check English through the paper carefully.
Response 9: Modifications have been made to the original text.

Round 2
Reviewer 1 Report
Comments and Suggestions for Authors
(1) Some of the concerns are not addressed.
For example, "Please present the comparison studies among the HLIP-s2s, standard-DDC-s2s and Robust-DDC-s2s in Section 5.2.2 and 5.2.3, as you did in Section 5.2.1." is not solved.
(2) Does the baseline 'HLIP-S2S ' use the same controller as the one in Ref [12]? Why, in your case, does it work so badly? Is it probably because you are using a robot with heavy upper links? More in-depth discussion should be added.
Ref. [12] Xiong, X.; Ames, A. 3-d underactuated bipedal walking via h-lip based gait synthesis and stepping stabilization. IEEE Transactions on Robotics 2022, 38, 2405–2425.
(3) Section 5.2.4 is good. But you should contain the sim2sim experiment in the video
(4) Please mark the changes in a different color. Otherwise, it is difficult to evaluate your work. For example, in line 81, the A and B are still not explained. Even though you responded to it in the letter.
Comments on the Quality of English Language
Can be easily understood.
Author Response
(1) Some of the concerns are not addressed.
For example, "Please present the comparison studies among the HLIP-s2s, standard-DDC-s2s and Robust-DDC-s2s in Section 5.2.2 and 5.2.3, as you did in Section 5.2.1." is not solved.
Response 1: Thank you for the reviewer’s suggestion. The reason we did not compare with standard-DDC-s2s in Section 5.2.2 and 5.2.3 is as follows: this paper focuses more on the introduction of the robust method, and the standard DDC method is not the primary emphasis.
(2) Does the baseline 'HLIP-S2S ' use the same controller as the one in Ref [12]? Why, in your case, does it work so badly? Is it probably because you are using a robot with heavy upper links? More in-depth discussion should be added.
Ref. [12] Xiong, X.; Ames, A. 3-d underactuated bipedal walking via h-lip based gait synthesis and stepping stabilization. IEEE Transactions on Robotics 2022, 38, 2405–2425.
Response 2: Thank you for the reviewer’s suggestion. I have added further explanation in Remark 3.
(3) Section 5.2.4 is good. But you should contain the sim2sim experiment in the video
Response 3: I have already added that part to the video.
(4) Please mark the changes in a different color. Otherwise, it is difficult to evaluate your work. For example, in line 81, the A and B are still not explained. Even though you responded to it in the letter.
Response 4: The latest revisions have been highlighted in red.
Reviewer 2 Report
Comments and Suggestions for Authors
1.While the authors employ robust data-driven control (RDDC), the paper would benefit from a more thorough comparison with other data-driven approaches such as reinforcement learning or supervised learning. Including a performance comparison would highlight the specific advantages of RDDC.
2.The experimental simulations primarily focus on payload variations, sloped terrains, and push recovery. The study could be strengthened by expanding to more dynamic environments, such as uneven terrain or obstacles, to further assess the robustness of the proposed controller.
3.The use of ellipsoid bounds for noise in the S2S dynamics is an interesting approach, but the justification for choosing these specific bounds is unclear. The paper should explain how these bounds were selected and whether they were fine-tuned during the experiments or based on prior knowledge.
4.The complexity of the robust data-driven controller and its computational demands should be discussed in the context of real-time implementation. Specifically, it would be useful to know if the system can maintain the same performance on physical hardware with limited computational resources.
5.The feedback gain calculation using noisy data matrices is novel, but it would be beneficial to explain how sensitive the feedback gain is to different levels of noise. Including a sensitivity analysis could clarify whether the feedback controller remains stable under varying noise conditions.
Comments on the Quality of English LanguageMinor editing of English language required.
Author Response
Comments 1:While the authors employ robust data-driven control (RDDC), the paper would benefit from a more thorough comparison with other data-driven approaches such as reinforcement learning or supervised learning. Including a performance comparison would highlight the specific advantages of RDDC.
Response 1:Thank you for the reviewer's comments. Currently, I have supplemented comparative experiments with other machine learning methods, namely Lasso least squares identification. Since the work presented in this paper focuses more on direct data-driven control, it is indeed necessary to supplement comparisons with traditional indirect data-driven control methods (i.e., parameter identification followed by controller design). However, the comparison with reinforcement learning has not been completed yet due to the lengthy training time required by reinforcement learning and the extensive engineering tuning needed for designing the reward function. I will attempt to explore this area in my subsequent work.
Comments 2:The experimental simulations primarily focus on payload variations, sloped terrains, and push recovery. The study could be strengthened by expanding to more dynamic environments, such as uneven terrain or obstacles, to further assess the robustness of the proposed controller.
Response 2:I have added test experiments for uneven terrain scenarios in Mujoco, which further validate the robustness of the current method. However, obstacle avoidance experiments have not been included because the current work primarily revolves around the Stepping Controller. For obstacle avoidance scenarios, the emphasis is more on global path planning (which requires the addition of a higher-level planner within the current control framework), and this is one of the areas I hope to explore in my future work. The current method has already addressed issues related to lateral and sagittal walking, as HLIP decouples movements in these two directions, and changes in the yaw direction can also be achieved accordingly. Therefore, it is expected that if a higher-level obstacle avoidance planner (such as a global path planner based on control barrier functions) provides the corresponding CoM position and velocity trajectory, the current RDDC-WBC can execute the commands accordingly.
Comments 3:The use of ellipsoid bounds for noise in the S2S dynamics is an interesting approach, but the justification for choosing these specific bounds is unclear. The paper should explain how these bounds were selected and whether they were fine-tuned during the experiments or based on prior knowledge.
Response 3: Thank you for the reviewer's comments. Supplementary explanations regarding the selection of parameters have now been provided in Remark3. This issue has a strong correlation with Issue 5. Theoretically, the larger the noise added, if Problem (16) remains feasible, the greater uncertainty the Stepping controller can stabilize. However, excessive noise can render Problem (16) infeasible, so in actual experiments, we chose the maximum noise that could guarantee the feasibility of Problem (16) as a parameter. Since data collected from simulators can be approximated as noise-free, the size of the added bounded energy ellipsoid noise can be regarded as the boundary of the maximum disturbance the system can experience during experiments. Meanwhile, despite numerous experiments conducted to demonstrate its robustness, for LIP-class models, the system's input-output only includes step length and CoM state. Therefore, in practice, perturbations to the system's state, whether due to thrust or load, are ultimately treated as perturbations to the centroid position and velocity (for thrust, it can also be seen as a sudden change in state over a very short period of time). Consequently, as the bounded energy ellipsoid increases, larger CoM perturbations can be overcome. However, quantifying a reasonable range of noise seems difficult, as it is challenging to find a reasonable mapping from the physical domain (i.e., equations reflecting changes in Newton's laws due to thrust and load variations) to the data domain. The field of uncertainty quantification seems to address this issue, but adding the upper bound of the uncertainty identified online to the SDP problem may render the problem infeasible. Therefore, this direction still requires further exploration to be completed.
Comments 4:The complexity of the robust data-driven controller and its computational demands should be discussed in the context of real-time implementation. Specifically, it would be useful to know if the system can maintain the same performance on physical hardware with limited computational resources.
Response 4: The current overall control framework involves solving the robust feedback control gain K offline and utilizing this gain online to implement a deadbeat-style state feedback controller. Therefore, the time required online is negligible, as the current framework does not solve an optimization problem for the online stepping controller. Within the entire hardware framework of the BRUCE robot, the stepping controller and Whole-Body Controller (WBC) run as separate threads, both at 500Hz, while the current Robust Data-Driven Controller (RDDC) can solve within 1000Hz. In potential future extensions, the robust feedback control gain K may need to be solved online (e.g., using online data to continuously update the control gain K). Even so, the current time to solve the problem once in CVXPY is less than 1ms, indicating that the framework has strong scalability (e.g., adding more constraints to the SDP problem). Additionally, in hardware implementation, C++ can be utilized to construct the problem for faster solution frequencies.
Comments 5:The feedback gain calculation using noisy data matrices is novel, but it would be beneficial to explain how sensitive the feedback gain is to different levels of noise. Including a sensitivity analysis could clarify whether the feedback controller remains stable under varying noise conditions.
Response 5: In Section 5.3 of the paper, a sensitivity analysis has been added.